# Decoupling of Nrf2 Expression Promotes Mesenchymal State Maintenance in Non-Small Cell Lung Cancer

**DOI:** 10.3390/cancers11101488

**Published:** 2019-10-02

**Authors:** John A. Haley, Christian F. Ruiz, Emily D. Montal, Daifeng Wang, John D. Haley, Geoffrey D. Girnun

**Affiliations:** 1Departments of Pathology, Stony Brook University School of Medicine, Stony Brook, NY 11794, USA; john.haley@umassmed.edu (J.A.H.); montale@mskcc.org (E.D.M.); geoffrey.girnun@stonybrookmedicine.edu (G.D.G.); 2Bioinformatics and Stony Brook Cancer Center, Stony Brook University School of Medicine, Stony Brook, NY 11794, USA; daifeng.wang@wisc.edu

**Keywords:** Nrf2, epithelial mesenchymal transition, glycolysis, TCA cycle, lipogenesis, redox signaling

## Abstract

Epithelial mesenchymal transition is a common mechanism leading to metastatic dissemination and cancer progression. In an effort to better understand this process we found an intersection of Nrf2/NLE2F2 (Nrf2), epithelial mesenchymal transition (EMT), and metabolic alterations using multiple in vitro and in vivo approaches. Nrf2 is a key transcription factor controlling the expression of redox regulators to establish cellular redox homeostasis. Nrf2 has been shown to exert both cancer inhibitory and stimulatory activities. Using multiple isogenic non-small cell lung cancer (NSCLC) cell lines, we observed a reduction of Nrf2 protein and activity in a prometastatic mesenchymal cell state and increased reactive oxygen species. Knockdown of Nrf2 promoted a mesenchymal phenotype and reduced glycolytic, TCA cycle and lipogenic output from both glucose and glutamine in the isogenic cell models; while overexpression of Nrf2 promoted a more epithelial phenotype and metabolic reactivation. In both Nrf2 knockout mice and in NSCLC patient samples, Nrf2^low^ was co-correlated with markedly decreased expression of glycolytic, lipogenic, and mesenchymal RNAs. Conversely, Nrf2^high^ was associated with partial mesenchymal epithelial transition and increased expression of metabolic RNAs. The impact of Nrf2 on epithelial and mesenchymal cancer cell states and metabolic output provide an additional context to Nrf2 function in cancer initiation and progression, with implications for therapeutic inhibition of Nrf2 in cancer treatment.

## 1. Introduction

The initiation and progression of adenocarcinomas of the lung alter networks important in cell stemness and differentiation state, in both epithelial and associated stroma. Developmental models demonstrate roles for mutant KRAS in the establishment of AT2 cell adenocarcinomas dependent on SOX2 and Notch temporal expression levels [1] and both SOX2 and SOX9 can promote cellular trans-differentiation via epithelial mesenchymal transition (EMT) [2,3]. 

The capacity of cancer cells to undergo epithelial mesenchymal transition (EMT) is strongly implicated as an important driver of tumor cell metastasis and of resistance to therapy [4,5]. This form of cell plasticity is associated with the acquisition of enhanced migratory/invasive cell programs and activation of anti-apoptotic and drug/radiation resistance programs [6,7,8,9,10]. The epigenetic reprogramming through EMT enables tumor cells to acquire new phenotypes and pathological behaviors [11,12] including disassembly of epithelial cell-junctions, loss of epithelial polarity, and formation of molecular assemblies that enable migration and invasion [13,14]. Importantly, migratory mesenchymal-like cells can undergo a reverse transition back to a more epithelial state through which cells regain proliferative potential and can establish metastases with phenotypic and mutational characteristics similar to the original primary tumor [15]. EMT-derived cells can exhibit stem-like properties [16] that include the ability to initiate tumors at low cell density in vivo, sphere formation in vitro, and expression of potential pluripotent stem cell associated markers (i.e., CD44^high^, CD24^low^, ALDH1) that likely reflect the underlying chromatin remodeling events. Distinctively, later stage lung tumors can exhibit nests of stable mesenchymal sarcomatoid-like cells [17] associated with aggressive proliferative and migratory behaviors. Current evidence shows that mesenchymal-like tumor cells can be positively selected for by chemotherapy in solid tumors [18] and resist apoptosis in part through suppression of BIM [19]. The effective therapeutic targeting of these reversible /metastable and sarcomatoid cancer states is an unmet medical need [20,21], in part due to the high degree of cellular heterogeneity [22] and the acquisition of parallel signaling pathways within individual cells.

Mutations in KRAS and EGFR are common driver mutations in lung adenocarcinomas, with ~25% and ~15% incidence, respectively. Cells harboring oncogenic mutations in KRAS or EGFR can undergo metastable EMT through the addition of growth modulators, for example, TGFβ [23,24] or oncostatin M [25], with subsequent activation of epigenetic reprogramming transcription factors notably Snail [26] followed by Zeb1 [27]. Initially EMT can be reversed through the withdrawal of inducer, but through mechanisms not well understood, cells can establish a stable mesenchymal state supporting both proliferation and migration programs. Both KRAS and EGFR activation can promote EMT [28,29] but once a mesenchymal state is established, previously oncogene addicted cells become resistant to KRAS knockdown or EGFR inhibition [30,31], findings which have therapeutic relevance [9,10,32,33]. The targeting of mesenchymal-like tumor cell states has proven elusive, in part due to the high degree of cellular heterogeneity [22] and the acquisition of parallel signaling pathways within individual cells. The mesenchymal-like cell state derived by EMT has been associated with altered autocrine and paracrine growth factor receptor signaling [34], stromal and inflammatory cell interaction [35], and metabolic output [36,37]. 

Cancer cells must change their metabolic landscape in order to sustain a proliferative state. However metabolic changes associated with the different stages of metastases are not well understood. Proliferating tumor cells can increase glycolytic, lipid, and nucleic acid synthesis through the Warburg effect. However, both decreased and increased glycolysis and oxidative phosphorylation have been correlated with cancer metastasis, seemingly in a model specific manner. Our recent studies of Zeb1, Snail, and TGFβ induced metastable EMT suggested a marked reduction in RNAs encoding genes that regulate the redox metabolism and glycolysis in the mesenchymal state [38]. Specifically, we observed the genes regulating the redox state are targets of the antioxidant transcription factor Nuclear factor erythroid-2 like-2 (Nrf2/NFE2L2). 

Growth factor activation and mutations in KRAS increase Nrf2 transcription, in part through AP1, and Nrf2 expression, which correlate with tumor viability, proliferation [39], and mRNA translation [40]. Nrf2 protein abundance is carefully regulated by ubiquitination through redox state sensitive E3 ligase complexes including KEAP1/CUL3, BTRC/SKP1, and SYVN1 [41], as well through DNA mutation [42] and through alternative RNA splicing. Nrf2 activating mutations in KEAP1, CUL3, or Nrf2 itself increase cell proliferation and tumor expansion in vivo [43,44] and can contribute to drug resistance [45]. In normal cells, Nrf2 can exert a geno-protective function by preventing excess reactive oxygen species (ROS), while in tumor cells Nrf2 activation contributes to tumorigenesis [39,43,44,46].

The role of Nrf2 functions in tumor initiation and tumor progression varies between different model systems, and its apparent contextual functions are not well defined [47]. Here we show a relationship between EMT, Nrf2, and metabolic reprogramming, using stable isotope carbon tracing based GC/MS, LC-MS/MS, and RNAseq in isogenic non-small cell lung cancer (NSCLC) adenocarcinoma models of epithelial and mesenchymal-like states. These isogenic NSCLC cell line models reflect clinically relevant activating mutations in EGFR and KRAS. We observe that in the mesenchymal state, both Nrf2 activity and Nrf2 protein abundance are attenuated. Metabolically, the mesenchymal state was characterized by decreased glycolysis, lipid synthesis, and TCA cycle intermediates relative to its isogenic epithelial counterpart. Experiments with gain of Nrf2 expression were associated with a shift to a more epithelial phenotype and associate marker changes. In contrast, knockdown of Nrf2 promoted a mesenchymal phenotype with reduced lipogenic and glycolytic function. The data support a model where the consequences of Nrf2 activity are in part dependent on epithelial or mesenchymal state epigenetic contexts, relevant to the potential therapeutic use of Nrf2 inhibitors or activators in cancer treatment or prevention.

## 2. Results

### 2.1. Molecular Characterization of Isogenic NSCLC EMT Cell States

EMT is highly associated with a drug resistant phenotype [5] and an altered apoptotic response [48]. We investigated the metabolic reprogramming associated with EMT in NSCLC adenocarcinoma models reflecting a reversible metastable EMT state (Figure 1A). Cell lines include mutant, activated EGFR (HCC4006 and HCC827), and mutant activated KRAS (H358 and A549). Paired isogenic comparisons of RNA, protein, and metabolite abundance (Appendix A) in these four lung cancer cell lines were used because of the high molecular heterogeneity in both epithelial and mesenchymal cell states (Appendix A). We have previously shown that with short term three- to seven-day induction of EMT, the transition is still proceeding at the single cell level [25]. Here mesenchymal states were induced for greater than twenty-one days to establish a stable mesenchymal state with transition with markers differentiating epithelial and mesenchymal states (Figure 1B) and loss of E-cadherin from the plasma membrane (Figure 1C). Analysis of RNAseq log2 fold changes between epithelial and mesenchymal states show expected alterations in lung development, metabolic and redox sensing gene expression. The development of lung tissues requires the interaction of epithelial and mesenchymal cell signaling components, involving WNT, SHH, BMP, and FGF pathways [49]. RNA expression datasets indicated progenitor markers GATA6 and ID2 were generally increased in the mesenchymal state, while SOX9 showed little change (Appendix A). Alveolar AT1 cell markers (AQP5, AQP3, and HOPX1) and ciliated cell marker FOXJ1 were decreased in the mesenchymal state, while podoplanin (PDPN) was not expressed. The endoderm differentiation marker, FOXP1, was increased, as were the stem niche factors WNT5A and WNT5B. Anticipated RNA and protein changes in epithelial and mesenchymal markers, and significant co-correlation with GSEA signatures characteristic of epithelial mesenchymal transition (Appendix A), were observed. Therefore, the four isogenic cell line models reflect changes associated with metastable partial EMT.

Interestingly, we found that compared to the epithelial state, mesenchymal-like cells had alterations in the levels of RNA in several metabolic pathways including glycolytic and pentose phosphate pathway (PPP) genes (Figure 1D). Proteomic data also support a reduction in glycolytic and PPP proteins G6PD, HK2, PFKFB2, and GPD2 proteins (data not shown). A similar reduction in TCA cycle and lipid synthesis RNAs were observed (Figure 1D). We previously observed similar Nrf2 target RNA changes with doxycycline-inducible TGFβ, Zeb1, and Snail in a H358/KRAS background [38], suggesting these findings are not restricted to TGFβ signaling.

### 2.2. Altered Glucose, Glycolysis, and TCA Cycle Metabolites Between Epithelial and Mesenchymal mtEGFR and mtKRAS Cell States

We sought to determine whether the decrease in glycolytic, lipid synthesis and TCA cycle RNA expression would reflect functional metabolic changes. Previous studies suggest that glycolysis can be increased [50,51,52] or decreased [53] with metastatic progression in NSCLC, possibly depending on the degree of the pro-migratory mesenchymal state and the pro-proliferative re-epithelialization associated with mesenchymal epithelial transition (MET). Therefore, we asked whether the change in glycolytic RNA expression (Figure 1D) was associated with functional changes in glycolysis. The HCC4006 and A549 models were maintained for three weeks in control (epithelial) or TGFβ containing (mesenchymal) media, followed by ^13^C6-glucose addition for the final sixteen hours and analyzed by GC-MS. We observed a significant reduction in extracellular m+3 lactate in the mesenchymal state in the A549 and HCC4006 cells suggesting a reduction in glycolysis (*p* < 0.001; Figure 1E, with isotopologue distributions in Appendix A). In addition, extracellular acidification rate (ECAR), a surrogate measure of glycolysis was significantly reduced (Figure 1F). We observed decreased ^13^C-labeled G6P and PEP by GC-MS (Figure 2A, with isotopologue data Appendix A). We also observed an increase in extracellular glucose (*p* < 0.01; Figure 1E), which is consistent with reduced HK2 RNA, protein, and G6P data, and suggesting that glucose entry into glycolysis is reduced. Overall these data demonstrate a reduction in glycolysis in the mesenchymal state. 

Decreased ^13^C enrichment into PPP metabolite R5P was observed in the mesenchymal state (Figure 2A; with isotopologue data Appendix A), along with decreased G6PD RNA expression by both RNAseq (Figure 1D) and RT-PCR (data not shown), suggesting that glucose carbons were not being shunted to the pentose phosphate pathway. Therefore, glycolysis is decreased following long term EMT induction and establishment of the mesenchymal phenotype, consistent with other EMT models [36,53]. 

The reduction in glycolysis prompted us to examine TCA cycle metabolites, to determine whether mesenchymal state cells compensate for reduced glycolytic output by increasing oxidative metabolism [53,54]. We measured TCA cycle intermediates in epithelial and mesenchymal states, using ^13^C6-glucose or ^13^C5-glutamine. Interestingly, mesenchymal state A549 and HCC4006 showed significantly reduced levels (*p* < 0.05) of multiple TCA cycle intermediates, including citrate, αKG, fumarate, and malate from both glucose (Figure 2C) and glutamine (Figure 2D, with isotopologue data in Appendix A). We measured oxygen consumption rate (OCR) to determine whether the reduction in TCA cycle intermediates resulted in reduced mitochondrial OxPhos. Basal OCR was reduced in mesenchymal state cells (Figure 2B). Taken together, these data suggest that mesenchymal state cells are metabolically quiescent, with a reduction in glucose utilization, glycolytic capacity, and basal mitochondrial respiration. 

### 2.3. Reduced Lipid Synthesis and Redox Signaling in the Mesenchymal Cell State 

More recent studies show that EMT represses FASN expression and lipogenesis [36]. Therefore, we measured the protein expression of FASN, ACLY, and ACC enzymes and found they were reduced in mesenchymal state cells (Figure 3A). Similarly, a decrease in the RNA expression for FAAH, FAAH2, FA2H, ECI1, and CH25H was observed (Figure 1D) as was the helix-loop-helix transcription factor sensor of cholesterol and fatty acid levels SREBF1/SREBP1 in A549, HCC4006 (Figure 3B), H358, and HCC827 cells (Appendix A). In addition, we used ^13^C6-labeled glucose, ^13^C5-glutamine, or ^13^C2-acetate to determine whether de novo lipogenesis from specific substrates is reduced or whether there is an overall reduction in lipogenesis. Our data show that de novo palmitate synthesis and acetyl-CoA enrichment, an indicator of de novo lipogenesis, from the three ^13^C-labeled substrates are reduced in the mesenchymal state in A549 and HCC4006 (Figure 3C; with isotopologue data Appendix A), and in H358 and HCC827 (Appendix A) reinforcing the previous studies showing a decrease in fatty acid synthesis [36]. While metabolic output generally decreases in the mesenchymal state lung cells, we did observe accumulation of carbamoyl phosphate and aspartate in the mesenchymal state, by targeted LC-MS/MS (Appendix A) and by GC-MS using ^13^C6-glucose or ^13^C5-glutamine (Appendix A). 

Gene ontology analysis of the RNA expression datasets suggested a significant decrease in redox signaling pathways (Figure 4A). Consequently, we measured metabolites utilized in reactive oxygen scavenging. Targeted LC-MS/MS measurement of glutathione (GSH) and NADP+ (Figure 4B) were decreased.

### 2.4. Nrf2 Activity and Protein Abundance Are Regulated by EMT State

The transcription factor Nrf2/NFE2L2 is a central regulator of cell redox state. Therefore, we asked whether Nrf2 activity was altered with our EMT model by examining the abundance of Nrf2 and Nrf2 transcription targets in the RNAseq datasets. Nrf2 redox target genes were markedly decreased by RNAseq illustrated in Figure 4C while RNA encoding Nrf2 itself was relatively unchanged (along with RT-PCR and RNAseq in Appendix A). These observations were consistent with the decreased Nrf2 target RNA abundance in doxycycline inducible H358-TGFβ, -Zeb1 and -Snail cell lines [38], suggesting that a decrease in Nrf2 target RNA expression is not confined to TGFβ signaling but is a more general phenotype associated with the mesenchymal state. Finally, a similar reduction of Nrf2 target proteins is observed by stable isotope labeled proteomic approaches (data not shown). 

While Nrf2 RNA is relatively unchanged between EMT states, the abundance of Nrf2 protein was decreased (Figure 4D). In response to decreasing cellular reactive oxygen (ROS) levels, Nrf2 protein is degraded by a thiol sensitive KEAP1-CUL3-RBX1 [55,56], BTRC-SKP1-CUL1-RBX1, and SYVN1 ubiquitin ligase complexes [41]. The decreased abundance of Nrf2 protein was associated with increased ROS (Figure 4E). To further investigate the regulation of Nrf2 levels in epithelial and mesenchymal states, we measured the stability of Nrf2 after both proteasome inhibition and ribosomal translation blockade. Nrf2 protein abundance was measured after proteasome inhibition with MG132 (10 µM) for six hours in epithelial and mesenchymal, A549 and HCC4006 backgrounds (Figure 4F). Control cells showed reduced Nrf2 in the mesenchymal state. By six hours, proteasomal inhibition resulted in a marked increase in Nrf2 protein to similar levels in both cell states. These data indicate that Nrf2 is degraded in part through a proteasomal pathway in both mt-KEAP1 (A549) and wt-KEAP1 (HCC4006). To determine the effect of the mesenchymal state on de novo Nrf2 protein synthesis we treated A549 and HCC4006 cells in both states with cycloheximide (10 µM) for one and three hours (Appendix A). Nrf2 protein was rapidly reduced in both epithelial and mesenchymal states and in mutant (A549) and wild type (HCC4006) KEAP1 backgrounds. The decrease in Nrf2 protein and Nrf2 activity and increase in ROS suggest EMT alters the stability of Nrf2, in part independent of KEAP1, and that the increase in ROS might be a result of decreased Nrf2. 

### 2.5. Decoupling of Nrf2 Regulation Facilitates the Mesenchymal-Like Lung Cell State

We sought to determine whether stable knockdown of Nrf2 by shRNA in epithelial state cells might recapitulate aspects of the decreased Nrf2 in the mesenchymal state including metabolic and mesenchymal phenotype. Knockdown of Nrf2 in epithelial state A549 and HCC4006 cells with two independent shRNAs reduced Nrf2 protein as expected (Figure 5A). Interestingly, Nrf2 knockdown decreased epithelial state proteins, E-cadherin and FASN, and increased mesenchymal state proteins, fibronectin and N-cadherin. MEK phosphorylation of Erk/MAPK on the activation loop TEY motif (pErk) is increased by EMT in the NSCLC models [57]. Here pErk is increased by Nrf2 knockdown. Metastable EMT typically reduces proliferation. Knockdown of Nrf2 decreased cell proliferation of A549 and HCC4006 cells (Figure 5B) consistent with previous findings [58]. EMT is characterized by the loss of E-cadherin and other junctional proteins from the cell membrane. Here knockdown of Nrf2 resulted in loss of membrane E-cadherin as evidenced by immunofluorescence (Figure 5C). We asked whether over-expression of Nrf2 would yield an opposite result and increase an epithelial phenotype. Over-expression of exogenous Nrf2 by adenovirus infection in HCC4006 markedly increased Nrf2 protein, decreased N-cadherin, and increased E-cadherin (Figure 5D), biomarkers of an epithelial state. Finally, we wanted to determine whether ROS were in part responsible for the observed effects of the mesenchymal state. We treated in HCC4006 cells with N-acetyl cysteine (NAC). NAC promoted a loss of mesenchymal markers fibronectin and N-cadherin and a gain of the epithelial marker E-cadherin (Figure 5E). 

We then performed 13C tracer studies in epithelial and mesenchymal state of Nrf2 knockdown cells. Similar to the metabolic state of the mesenchymal phenotype, Nrf2 stable knockdown significantly decreased G6P, PEP, lactate, palmitate production (Figure 6A,D), decreased extracellular glucose consumption (Figure 6B), and increased aspartate abundance by GC-MS using ^13^C6-glucose as a tracer (Figure 6C). Conversely, overexpression of Nrf2 in mesenchymal state cells significantly promoted glucose derived ^13^C incorporation (*p* < 0.01) into palmitate (Figure 6E; isotopologue distributions are shown in Appendix A). Using RNA, protein and targeted metabolomics, these studies show that a decrease in Nrf2 is promoting the mesenchymal state phenotype. 

### 2.6. Increased Mesenchymal Alveolar Cell Characteristics in Nrf2 Knockout Mice

We examined a previously published microarray data set from mouse lung alveolar cells with Nrf2 knockout mice (Nrf2^−/−^) for epithelial and mesenchymal cell characteristics. Alveolar type II cells from Nrf2^−/−^ mice were previously compared with normal controls by microarray [59]. GSEA analysis showed a significant correlation with EMT related signatures (Figure 7A). We examined Nrf2^−/−^ data for characteristic markers of epithelial and mesenchymal cell states and correlated these with metabolic and Nrf2 target gene expression described above. Specifically, we correlated significant RNA changes in Nrf2^−/−^ alveolar cells with RNA changes in all four isogenic lung EMT state models and mean RNA changes in our previous H358 Zeb1-Snail dox-inducible models (Figure 7B). Overlap of Nrf2^−/−^ with EMT state markers (e.g., EHF, ELF3, JUP, LOX, SERPINE1) and EMT modulated metabolic markers (BCAT1, FA2H, and LRP5) were observed. Nrf2^−/−^ lung cells showed correlation with overlapping quartile RNAs in the individual four NSCLC isogenic epithelial and mesenchymal models, with regression values ranging from 0.74 to 0.86 (Figure 7C). The comparison with our previous doxycycline inducible Zeb1 and Snail models was performed to rule out TGFβ specific effects over more generalized EMT effects. The data suggested that Nrf2^−/−^ mouse alveolar cells display greater mesenchymal characteristics compared with wt-Nrf2 alveolar cells, further supporting the role for Nrf2 in EMT in lung cancer.

### 2.7. Human NSCLC TCGA Link Nrf2 Target Signatures to EMT Signatures and Metabolic Alterations

Our data shows that Nrf2 activity, epithelial phenotype, glycolytic and lipogenic markers are positively correlated in vitro and in vivo in Nrf2^−/−^ mice. We investigated whether this would be relevant in human lung adenocarcinoma patient specimens. First, we determined whether the Nrf2 target gene signature would co-correlate with common EMT markers. The nine Nrf2 target genes (Nrf2 sig: NQO1, GSR, GPX2, GSTA4, PRDX5, CBR1, GLRX, MGST1, GCLC) were altered in both the TGFβ induced EMT models (Figure 4C, Appendix A), and H358/Zeb1 and H358/Snail models [38]. Coordinate RNA expression of Nrf2 target genes, with EMT and metabolic genes (Figure 8A, RNAseq) as well as protein markers (Figure 8B) were observed. The Nrf2 signature RNAs defining Nrf2 inactive vs. Nrf2 active states (ΔNrf2) is shown (Figure 8C). 

We further validated the co-correlation between Nrf2 signature, metabolic changes, and EMT using an 85 gene EMT reference signature from Heymach and coworkers [60]. The Nrf2 target RNA signature was upregulated in ~23% of cases in The Cancer Genome Atlas (TCGA) lung adenocarcinoma provisional patient set (n = 503). As expected, the Nrf2 signature significantly correlated with KEAP1 mutation (q = 2.4 ×10^−4^; data not shown). A significant Pearson correlation cutoff (q < 0.05) between Nrf2 signature and CDH1 or VIM co-correlated RNA changes was observed (Appendix A). Regression analysis shows a significant correlation between Nrf2 target genes and EMT markers in the TCGA dataset (Appendix A). A similar Nrf2 correlation with CDH1 or VIM correlated RNA changes was performed in a smaller TCGA lung adenocarcinoma 203 patient set (Appendix A). Similarly, a significant correlation between the Nrf2 signature correlated TCGA genes and the EMT model RNA changes was observed (*p* < 0.01; Appendix A). 

We asked the reverse question, whether EMT related signatures would correlate with the expression of metabolic genes and Nrf2 target RNA expression in lung adenocarcinoma patient samples, similar to that observed in the lung model systems. A mesenchymal signature (VIM, FN1, LOX, GPC6, ZEB1) was defined as the highest co-correlated mesenchymal marker set in Cancer Cell Line Encyclopedia (CCLE) NSCLC cell lines. RNA expression in lung adenocarcinoma TCGA samples was co-correlated with the mesenchymal signature. Metabolic and lung development RNA indicative of a more de-differentiated mesenchymal-like state, changes were co-correlated with significant q-values (Appendix A) in two patient datasets. These changes included decreased lipogenic (SREBF1, FAAH, FA2H) and Nrf2 target genes (GPX2, PRDX5, CBR1), genes that were decreased in the mesenchymal state lung EMT models. Finally, in human lung adenocarcinoma samples, we observed co-correlations between mesenchymal markers, decreased Nrf2 function, and decreased RNAs related to lipid metabolism.

## 3. Discussion

Redox signaling has been implicated in cancer progression, where specifically Nrf2 has been shown to exert both tumor suppressive and tumor promoting activities [40,41,45]. However, studies on the role of Nrf2 in tumor cell behavior have not directly addressed the roles of cell plasticity and interconversion of epithelial and mesenchymal/neuroendocrine cell states. Here we sought to clarify the functions of Nrf2 in cancer progression using non-small cell lung models with isogenic epithelial and mesenchymal cell states. 

Nrf2 function in NSCLC models was shown be contextual. Nrf2 activity is markedly increased in a pro-proliferative, pro-metabolic epithelial cell state, while markedly decreased in the mesenchymal cell state despite increased ROS. We showed that knockdown of Nrf2 promoted the mesenchymal cell state and attenuated metabolism. In contrast, overexpression of Nrf2 enhanced an epithelial phenotype with increased markers of lipogenesis, oxidative phosphorylation, and glycolysis. The correlations between EMT cell state, intermediary metabolic output, and Nrf2 activity are significant in both Nrf2 knockout mouse model and in human NSCLC adenocarcinomas, with implications for the use of Nrf2/redox modulators in cancer therapy. 

### 3.1. Nrf2 Activity Favors the Epithelial Tumor Phenotype

Reactive oxygen is increased by growth factor signaling and oncogenic mutation, associated with increased glycolytic output. In both KRAS driven and EGFR driven NSCLC models, we observe that Nrf2 protein abundance (Figure 4D) and activity (Figure 4A,C and Appendix A) are markedly altered between epithelial and mesenchymal cell phenotypes. In the mesenchymal state we show a decrease in RNA abundance of Nrf2 transcription targets which are important to the regulation of cellular redox homeostasis. Despite reduced glycolytic, TCA cycle and lipogenic metabolic output in the mesenchymal state (Figure 2 and Figure 3, and Appendix A) a significant accumulation of ROS is seen by flow cytometry (Figure 4E). We also observed alterations in markers of oxidative stress by LC-MS/MS (Figure 4B). This is consistent with a model that ROS production is needed for EMT initiation [61] as well as for the maintenance of a mesenchymal cancer cell state. We observe that Nrf2 protein levels are regulated by protein turnover [55,56] (Figure 4F and Appendix A) in both epithelial and mesenchymal states. While the KEAP1 mutation in A549 cells prevents E3 ligase mediated ubiquitination and proteasomal degradation of Nrf2, the Nrf2 protein was more unstable in both mutant (A549) and wild type (HCC4406) KEAP1 backgrounds in the mesenchymal state (Figure 4F and Appendix A). Interestingly, in the A549 mesenchymal state Nrf2 is still reduced in abundance suggesting non-KEAP1 mechanisms are responsible for these effects [62]. An increase in mesenchymal state ROS can be derived from electron leakage involving mitochondrial Complex I and Complex III. However, mitochondrial respiration is decreased so the reduction in Nrf2 protein abundance and activity is likely a dominant factor in the maintenance of ROS levels and a mesenchymal-like state. Typically, when ROS increases, KEAP1 function is disrupted allowing stabilization of Nrf2. Interestingly, we observed decreased Nrf2 protein in both KEAP1-wildtype HCC4006 and KEAP1-mutant A549 cell lines, indicating that EMT effects on KEAP1 function were not solely responsible for the decrease in Nrf2 protein. This is a point of future investigation, examining alternative splice forms of Nrf2, alternative ubiquitin E3 ligases which might recognize Nrf2 as substrate and translational control of Nrf2 synthesis. 

The role for Nrf2 expression in cancer progression has yielded divergent findings in different model systems. For example, in fibrotic lung disease, Nrf2 expression decreased the EMT promoting transcription factor Snail [63]. In contrast, in NSCLC H1975 and bladder RT4 cells Nrf2 knockdown had mixed impacts on epithelial and mesenchymal markers, suggesting the generation of an intermediate ‘hybrid’ EMT state [64]. Here in mutant KRAS and mutant EGFR NSCLC models we find Nrf2 expression promotes a more epithelial cell state while antagonizing the mesenchymal state. Nrf2 is known to play a dual role in cancer, where Nrf2 is understood to reduce genotoxic damage in normal cells, yet promote a proliferative behavior in established epithelial-derived carcinoma cells. Here we uncover different functions of Nrf2 in carcinoma dependent on epithelial or mesenchymal EMT cell state, a third role.

### 3.2. Nrf2, Glycolytic, TCA, and Lipid Metabolic Pathways Are Coordinately Attenuated in the Mesenchymal-Like Carcinoma State

An increase in glycolysis and a reduction in oxidative phosphorylation historically has been associated with neoplastic transformation and uncontrolled cell proliferation through the Warburg effect. The controversy around increased versus decreased glycolytic, TCA cycle and lipogenic output in promoting metastatic dissemination, not only in lung but in solid tumors in general, suggests these events are highly contextual. We show correlation of Nrf2 with an epithelial tumor state. Nrf2 knockdown in epithelial state cells significantly reduces glycolytic output as measured by ^13^C6-glucose labelling of G6P, PEP, and lactate (Figure 6A and Appendix A). This was co-correlated with general decreases in multiple glycolytic and PPP enzymes in the NSCLC mesenchymal state, including phosphofructokinase (PFKFB2,3), fructose-1,6-biphosphatase (FBP1), hexokinase-2 (HK2), and glycerol 6-phosphate dehydrogenase (G6PD) at the RNA (Figure 1D) and protein levels (data not shown). Lipogenesis was reduced with Nrf2 knockdown, where decreased ^13^C6-glucose incorporation into palmitate (Figure 6D and Appendix A) was observed. Conversely, overexpression of Nrf2 increased de novo synthesis of palmitate from glucose (Figure 6E). These data are consistent with Boothman and coworkers showing a decrease in fatty acid synthesis and glycolysis with EMT in A549 cells [36]. Similarly, mutant EGFR HCC827 erlotinib resistant cells show EMT marker changes, decreased glycolysis and decreased TCA cycle intermediates [65], consistent with our findings. The significant co-correlation between Nrf2 activity and EMT markers, comparing wildtype and conditional Nrf2 knockout mouse alveolar cells (Figure 7A–C), is observed. Importantly, a significant co-correlation between Nrf2 activity, EMT and glycolytic, TCA cycle, and lipogenic metabolism is seen in human lung adenocarcinoma patient samples (Figure 8). This supports translation of Nrf2-EMT-metabolic signaling from isogenic cell models, to conditional knockout mouse models, to human lung patient specimens.

The activation of the Nrf2 pathway can promote glutamine uptake and glutathione synthesis through glutamate conversion [66]. Comparison of ^13^C5-glutamine and ^13^C6-glucose incorporation studies show that glutamine is not being utilized as an alternative fuel in the attenuated mesenchymal state TCA cycle or lipogenic output. However, aspartate does increase in the mesenchymal state using either labelled glucose or glutamine. The increase of aspartate from both ^13^C6-glucose and ^13^C5-glutamine in conjunction with increased carbamoyl phosphate and CPS1, could be acting as a possible non-toxic nitrogen “dump”. This, however, would require further investigation using heavy labeled nitrogen sources such as glutamine. 

As epithelial tumor cells undergo mesenchymal-like trans-differentiation, cell cycling time typically increases along with the acquisition of cell motility and invasion. The reduction in cell proliferation decreases the requirement for lipid and nucleic acid synthetic pathway activation, consistent with our findings in the four isogenic NSCLC models. However, the acquisition of ATP consuming myosin dependent motors of cell migration in the mesenchymal state suggests oxidative phosphorylation may need to be maintained during extensive migration possibly not observed in vitro. With that caveat, we suggest the association of metastasis with increased metabolic output, derives from re-epithelialization through MET and the formation of proliferating macro-metastases. 

### 3.3. Future Directions

The contextual relationship between Nrf2 activity and epithelial and mesenchymal cell states has therapeutic implications. Inhibitors of Nrf2 function [67] have been proposed in tumors with high Nrf2, for example, harboring inactivating mutations in KEAP1 or CUL3, or expressing active Nrf2 splicing isoforms. KEAP1 mutations, which typically decrease ubiquitylation and turnover of Nrf2, are found in 16.8% (154/915) of recent lung adenocarcinoma samples (MSK-IMPACT) [68], while CUL3 mutations are infrequent at 1.5% (14/915). Activating Nrf2 mutations are common in lung squamous carcinomas (6.2%; 71/1144) but are infrequent in lung adenocarcinomas (1.9%; 17/915), where Nrf2 activating driver mutations are only ~60% (17/915) of these mutations. In lung adenocarcinomas there is a trend toward co-mutation of KRAS and KEAP1. Given that Nrf2 can serve different roles dependent on epithelial or mesenchymal carcinoma cell contexts and that Nrf2 can protect from environmental carcinogenic impacts, acute Nrf2 inhibitor treatment strategies need to be considered to avoid promoting EMT in the first instance or increasing risk of tumor initiation in the second. Our observations that overexpression of Nrf2 promotes a more proliferative epithelial phenotype while conversely knockdown of Nrf2 promotes a more quiescent mesenchymal-like state, relevant to possible therapeutic intervention of Nrf2 signaling where EMT state would be predicted to impact outcome.

The transition from epithelial to mesenchymal-like states is relatively slow, taking place over days to weeks for full transition. The epigenetic reprogramming involved in EMT requires acetyl-CoA production for the histone acetylation changes. The sequence of events in attenuated metabolic output, in decreased Nrf2 activity, and in increased ROS production in the early initiation of EMT or in the later maintenance of a mesenchymal state are not well studied. If metabolic changes are necessary early events in EMT, then metabolic targeting may provide a means to reduce or prevent mesenchymal transition and tumor cell dissemination. The timing and coordination of early metabolic changes during EMT and their interdependencies are important points of future studies.

## 4. Experimental Procedures

### 4.1. Cell Culture

A549, H358, HCC827, and HCC4006 cell lines were grown in RPMI 1640 medium (GIBCO), containing 10% FBS (fetal bovine serum) at 37 °C and 5% CO2. Mesenchymal state cells had 10 ng/mL of TGFβ added to medium. Importantly TGFβ was maintained in the culture media throughout all experiments, to ensure a consistent epigenetic and metabolic state. Nrf2 knockdown were selected with 2 µg/mL puromycin. Cells were monitored by mycoplasma monthly by PCR [69]. All cells were purchased from the American Type Culture Collection (ATCC). 

### 4.2. Cell Proliferation Assay

Cells were plated in a 96 well plate. Seventy-two hours later the cells were fixed in 4% paraformaldehyde and stained with 0.5% crystal violet and absorbance at 570 nm was measured. 

### 4.3. Immunoblot

A549, H358, HCC827, and HCC4006 were immunoblotted using the following antibodies: Nrf2 (Santa Cruz #sc-13032), E-cadherin (BD #610181), N-cadherin (BD #610920), fibronectin (BD #610077), vimentin (Cell signaling #5741), beta-actin loading control (Sigma #A1978), FASN (Cell signaling #3180), ACC (Cell signaling #3662), ACLY (Cell signaling #4332) and p-Erk (Cell signaling #4370). Full immunoblots are shown in Appendix A. Densitometry ratios were calculated by Fiji (ImageJ) and normalized to β-actin. 

### 4.4. Immunofluorescence 

Cells were plated on coverslips and fixed with 4% paraformaldehyde. They were then permeabilized with 0.1% TritonX-100 and blocked in 5% BSA. Primary was done 1:50 in 5% BSA with E-cadherin antibody from Abcam (ab1416) for 1 hour. Secondary was done at 1:500 with Alexa Fluor 555 goat anti-mouse IgG from molecular probes (A21422) for 1 hour. The cover slips were then mounted on slides with Vectashield medium with DAPI from Vector laboratories (H-1200). The slides were then imaged with an Olympus Fluoview FV1000 confocal microscope. Slides were observed using a 60x Plan Apo N 1.42 oil immersion objective.

### 4.5. ^13^C-glucose, -glutamine and -acetate Tracer Studies

Cells were seeded in quintuplet into 60 mm dishes and allowed to grow overnight. [U2-^13^C2] sodium acetate, [U5-^13^C5]-glutamine, and [U6-^13^C6]-glucose (>99% purity and 99% isotope enrichment for each carbon position; Cambridge Isotope Labs) were used as tracers since they provide excellent analysis of overall central carbon metabolism and including the TCA cycle and lipogenesis [70]. Cells were incubated for 16 h with the respective ^13^C tracer. Briefly, following glucose, glutamine, or acetate treatment, culture medium was collected, and cells were washed in 0.9% NaCl, after which cell pellets were harvested. Specific extractions and analysis were performed as previously described [71,72,73,74,75] and below. 

TCA cycle intermediates and glycolytic intermediates were determined via an alternate method different to that of lipids. Cells were cultured in either media containing 2 mM ^12^C glutamine and 10 mM ^13^C6-labeled glucose, full media with 1 mM ^13^C2=labeled acetate, or 2 mM ^13^C5-labeled glutamine and 10 mM ^12^C glucose. Cells were washed and then scraped into 50% methanol:water, snap frozen three times, spun down and supernatant isolated for polar extraction. The supernatant was dried down and methoximated and derivatized with BSTFA. Citrate was monitored at 465–471, aspartate at m/z 334–338, fumarate at m/z 245–249, malate at m/z 335–339, glutamine at m/z 258–263, glutamate at m/z 348–353, lactate at m/z 219–222, αketoglutarate at m/z 304–309, G6P at m/z 357–359, PEP at m/z 369–372, R5P at m/z 357–359. 

For extracellular polar metabolites (glucose, glutamine, lactate, etc.), the harvested media was mixed 1:1 with 100% methanol and dried down. Samples were then methoximated and derivatized with BSTFA. The same m/z selection parameters were used. 

For lipids, the pellet was resuspended in 2:1 chloroform:methanol, and phase separated with 0.9% NaCl. The nonpolar phase was saponified and methylated with BF_3_ for an hour at 95 degrees Celsius. Palmitate was monitored at m/z 270-286, stearate at m/z 298-316, and myristate at m/z 242-256. 

Mass spectral data were obtained on the HP5973 mass selective detector connected to an HP6890 gas chromatograph. The settings are as follows: GC inlet 230 °C, transfer line 280 °C, MS source 230 °C MS Quad 150 °C. An HP-5MS capillary column (30 m length, 250 µm diameter, 0.25 µm film thickness) was used for both polar and lipid analysis. ^13^C6-glucose and ^13^C5-glutamine incorporation for measurement of glycolytic and TCA cycle intermediates was expressed as relative ^13^C abundance (Figure 2 and Figure 6, and Appendix A) with isotopologue maps shown in Appendix A. Incorporation of ^13^C6-glucose, ^13^C5-glutamine, and ^13^C2-acetate into palmitate was expressed as percent ^13^C-label and as acetyl CoA enrichment (Figure 3C and Figure 6D,E and Appendix A) with isotopologue maps in Appendix A. 

Agilent MassHunter MS Quantitative Analysis software was used for statistical analysis. ISOCOR was used to normalize data to an internal standard. For studies showing relative abundance of ^13^C labeled metabolites (and their isotopologues), 10 nM adonitol was used as an internal standard for the polar extractions and 10 nM heptadecanoic acid was used for lipid extractions. 

### 4.6. ECAR and OCAR SeaHorse Measurements

Cellular mitochondrial respiration rate and extracellular acidification rate, represented as OCAR and ECAR respectively, were measured using a Seahorse XFe96 Analyzer (Agilent). Cells were plated on a Seahorse 96 well plate and incubated for 16 h and then assayed with mitochondrial stress test kit (Agilent # 103015-100) or glycolytic stress test kit (Agilent #103020-100). 

### 4.7. LC-MS/MS Metabolomic Measurements

Metabolism data was obtained for isogenic epithelial and mesenchymal states. For each condition a minimum of six, 6 cm dishes of cells were cultured and ~70% confluence were used for metabolites extraction: After fast media aspiration, cells were lysed in 1.5 ml of 80% methanol in water (vol/vol, cooled to −80 °C) and incubated for ~30 min at −80 °C. Scraped cells were centrifuged for 5 min at 4 °C and 14,000 g. Supernatant from each plate was divided into three vials and dried in SpeedVac without heat. Water was added to each vial right before analysis based on cells count in the condition to inject metabolites from 2 × 10^5^ cells per run.

Metabolite separation, identification, and relative quantitation was performed using Agilent Technologies 1290 Infinity Binary LC System and 6490 Triple Quadrupole mass spectrometer. HPLC-MS conditions were: Heater for Amide XBridge HPLC column (3.5 μm; 4.6 mm × 100 mm, Waters) with the guard column was set to 30 °C. Mobile phase flow rate was 400 µl/min with solvent B changing from 85% to 30% in 3 min, then to 2% in next 9 min. After 3 min of isocratic flow, solvent B was changed back to 85% in 1 min, followed by 7 min of isocratic flow for column conditioning prior to the next injection. 

MS1 and MS2 resolution was set to Unit, with dwell time 5 ms. Source conditions were: drying gas 14 l/min at 200 °C, sheath gas 12 l/min at 300 °C, nebulizer gas 35 psi; Capillary voltage 4000 V and 3000 V, nozzle voltage was 0 V and 1500 V, High Pressure RF: 150 V and 90 V, Low Pressure RF: 60 V and 60 V in positive and negative mode, respectively. Data were acquired from 2 to 12 min using Agilent Technologies Mass Hunter Workstation LC/MS Data Acquisition for 6400Series Triple Quadrupole Software Version B.08.00 and processed using Agilent Technologies Mass Hunter Workstation Quantitative Analysis for Triple Quadrupole Software Version B.07.01 SP1. Statistical analysis of the quantitative data was done using Mass Profiler Professional (MPP) Version 14.5-Build 2772. 

### 4.8. RNA Transcript and Proteomic Analysis

Illumina HiSeq2000 RNA sequencing of duplicate H358 (GSE125365) and A549 (GSE125369) epithelial and mesenchymal states was deposited in GEO [57]. HCC827 (PRJNA203863), HCC4006 (PRJNA203863) FASTQ files were from Jia and coworkers [76]. H358, A549, HCC4006, and HCC827 epithelial and mesenchymal FASTQ file reads passing Illumina purity filter were aligned using TopHat2 and Cufflinks, with statistical analysis performed by CuffDiff, generating files of normalized counts for detected genes and transcripts (UCSC hg38). Galaxy server [77] was used for FASTQ alignment and analysis. Aligned RNAs passing QC thresholds were used to calculate mesenchymal–epithelial transcript abundance ratios followed by log2 linear scaling. H358-doxZeb1, H358-doxSnail, and H358doxTFGbeta Affymetrix U133-2 data (GSE125113) were previously described [38]. Mouse Nrf2 knockout alveolar cell microarray RNA data [59] with fold change *p* values <0.05 was used in co-correlation analyses. For comparison of RNAseq and microarray datasets, percent maximal value (% max; Figure 7B) was used to adjust for dynamic range variation.

To establish protein and phosphopeptide changes associated with metabolic networks, we used ^13^C-LysC6/ArgC6 SILAC labelled cells (H358, A549, and HCC4006) or iTRAQ labelled protein/phosphopeptide fractions (HCC827) from TGFβ induced epithelial and mesenchymal isogenic states. The use of isogenic matched models minimizes genotypic differences between cell lines derived from different cancer patients. Phosphopeptides were isolated by TiO2 affinity as previously described [57]. Protein identifications comply with established guidelines [78]. False discovery rates for peptide assignments ranged between 0.8% and 1.4%. 

### 4.9. RT-PCR

For RT-PCR, total RNA was extracted from cells using TRIzol. The reverse transcription reaction was performed using high capacity cDNA synthesis kit (Applied Biosystems). Quantitative real time RT-PCR analyses were performed using SYBRgreen on a QuantStudio 7 Flex instrument both from Applied Biosystems. Human 18S was used for normalization. 

### 4.10. Bioinformatics and Correlations

Nrf2 target gene co-correlations were performed using c-BioPortal and anchor genes NQO1, GSR, GPX2, GSTA4, PRDX5, CBR1, GLRX, MGST1, and GCLC. The difference between Nrf2 inactive and Nrf2 active states (ΔNrf2) is used in co-correlation analysis of TCGA lung adenocarcinoma datasets comprising a 503 RNAseq patient dataset (Figure 8 and Appendix A) or a 203-patient dataset (Appendix A) were queried. A mesenchymal signature (FN1, VIM, LOX, ZEB1, and GPC6 [ΔM-sig]; Appendix A) was correlated to the two TCGA lung adenocarcinoma datasets. Regression analysis and Pearson correlation (Figure 7C, Appendix A) were performed using R or JMPv10 (SAS). Gene set enrichment (GSEA) [79] was performed using GenePattern and the Broad Institute server (http://software.broadinstitute.org) (Figure 7A and Appendix A). Venn diagrams were constructed using (http://bioinformatics.psb.ugent.be). Changes in RNA, protein, and metabolites compare the mesenchymal-like state to the epithelial state (M/E), expressed as log2 fold change.

### 4.11. Measurement of ROS

Cells were plated in triplicate and left overnight. They were then suspended in HBSS containing 5µM CM-H2DCFDA (Thermofisher Scientific #C6827) for 30 minutes. Cells were measured for ROS activity on a FACScan Calibur using a 488 nm laser. ROS also was measured by LC-MS/MS by comparing oxidized to reduced glutathione (GSSG/GSH) ratios. 

### 4.12. Nrf2 Adenovirus

Nrf2 gene was cloned out of a pCDNA3-Myc3-Nrf2 (Addgene #21555) and into the pAdTrack-CMV vector. The AdTrack-Nrf2 construct was linearized and transformed into BJ5183-AD-1 Electroporation Competent Cells from Agilent (#200157). Plasmid DNA was linearized and transfected into AD293 cells using calcium phosphate. Nrf2-adenovirus went through multiple stages of amplification. Infected Nrf2 overexpressing cells were assayed within 48 h.

### 4.13. Nrf2 shRNA Lentivirus

Six shNrf2 constructs were purchased from Dharmacon. All were tested for Nrf2 knockdown and shNrf2 1, 2, and 3 were chosen on percent target knockdown. They had the following respective sequences: ATGAGTTCACTGTCAACTG, AGCATGCTGAAAACTTCGA, TTTTCTGCAATTCTGAGCA. For A549, shNrf2-1 and-2 were chosen due to better knockdown; while in HCC4006, shNrf2-2 and-3 were chosen. For all studies pools of stable puromycin resistant cells were used 2–3 weeks after infection and selection, with >500 individual clones per pool.

## 5. Conclusions

Migratory tumor cells can exist in a quiescent mesenchymal state, with the potential for re-epithelialization, re-acquisition of a proliferative phenotype, and metastatic outgrowth [15,22]. There is considerable debate regarding cell dependency on glycolysis and TCA cycle metabolism with cancer progression. Here, we find that Nrf2 activity and metabolism of metastatic lung cancer cells is highly dependent on their epithelial or reversible (metastable) mesenchymal cell states. Specifically, we find that mesenchymal-like lung adenocarcinoma cells coordinately decrease Nrf2, glycolytic, TCA cycle, and lipogenic activities relative to their isogenic epithelial counterparts, in vitro and in vivo. This is consistent with the attenuated proliferation, increased migration, and increased senescent behaviors of mesenchymal-like tumor cells. Analysis of breast cancer patient samples for high Nrf2 expression has correlated with a significant decrease in overall survival (*p* = 0.024) [64]. Here in NSCLC adenocarcinomas we observed a trend for poor overall survival with increased Nrf2 target gene expression (*p* = 0.15), which is correlated with a proliferative, epithelial phenotype. Taken together, these data support a model (Figure 8D) where EMT promotes dissemination of slowly proliferating migratory cells with low rates of glycolytic and lipogenic output, where the reestablishment of macro metastases at distant sites requires MET, and an epithelial proliferative phenotype with increased metabolic output. 

## Figures and Tables

**Figure 1 cancers-11-01488-f001:**
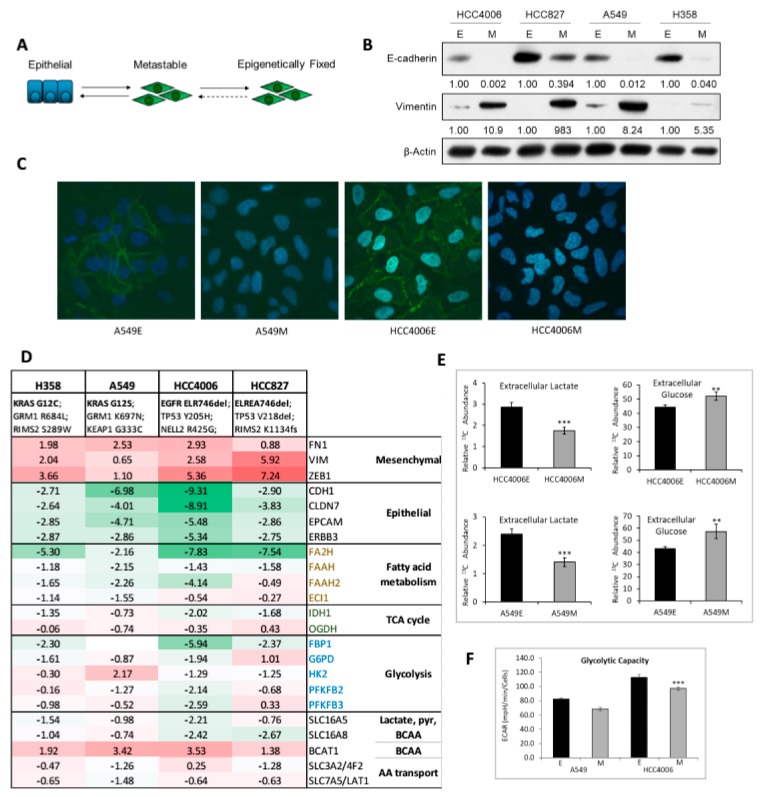
Isogenic mutant KRAS and EGFR model systems show altered metabolic RNA expression and attenuated glycolytic capacity when in a mesenchymal-like cell state. (**A**) Epithelial tumor cells can trans-differentiate to reversible/metastable and epigenetically ‘fixed’ mesenchymal cell states. (**B**) NSCLC isogenic mutant KRAS and EGFR models display epithelial or mesenchymal protein markers dependent on state. (**C**) Plasma membrane localization of E-cadherin (green) is lost in the mesenchymal state. Nuclei are stained with DAPI (blue). (**D**) RNA abundance characteristics of the four isogenic lung E-and M-state models (log2M/E fold change; bold values indicate FDR adjusted q value < 0.05). Selected metabolic RNA changes associated with M and E cell state. Glycolytic and pentose phosphate pathway M state RNA expression decreases, included phosphofructokinase PFKFB2/3), hexokinase-2(HK2), and glycerol 6-phosphate dehydrogenase (G6PD). TCA cycle IDH1 and OGDH, and lipid synthesis FA2H, FAAH, FAAH2, and ECI1 also were reduced in M state cells. (**E**) In HCC4006 and A549 labeled with 13C6-glucose, extracellular lactate and glucose levels decrease in M state by GC-MS. Isotopologue distributions are shown in Appendix A. (*p* < 0.05 *; *p* < 0.01 **; *p* < 0.001 ***). (**F**) In HCC4006 and A549 glycolytic capacity decreases in the M state.

**Figure 2 cancers-11-01488-f002:**
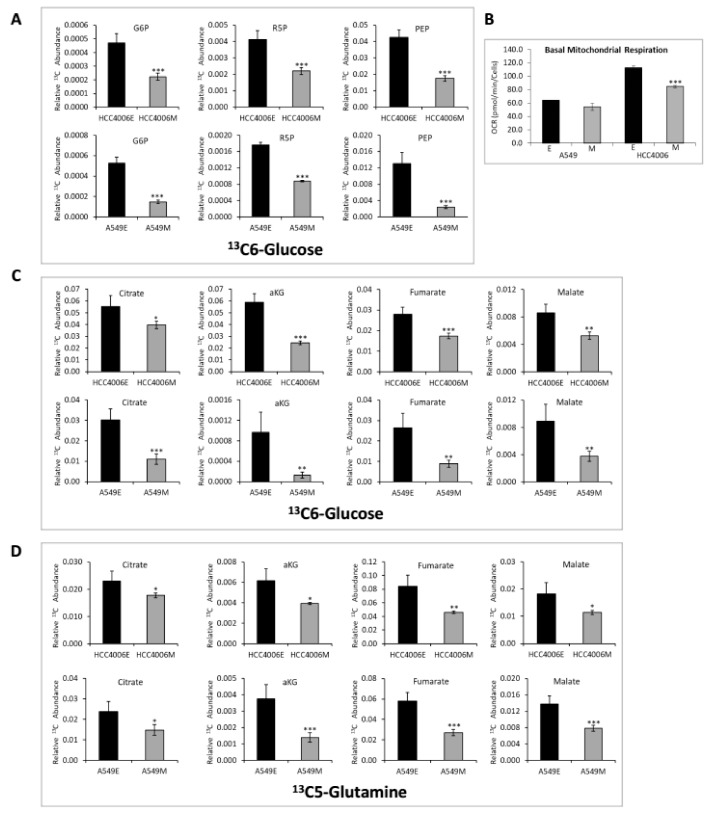
Reduced glycolytic and TCA cycle activity in the mesenchymal cell states. (**A**) In A549 and HCC4006 E and M state cells treated with 13C glucose, there is a decrease in glucose labeled glycolytic and pentose phosphate pathway metabolites in the M state. (*p* < 0.05 *; *p* < 0.01 **; *p* < 0.001 ***). (**B**) In A549 and HCC4006, basal mitochondrial respiration is reduced in M state cells. (**C**) In A549 and HCC4006 M state cells treated with 13C6-glucose, there is a decrease in glucose labeled TCA cycle metabolites. (**D**) In A549 and HCC4006 M state cells treated with 13C5-glutamine, there is a decrease in glutamine labeled TCA cycle metabolites. Isotopologue distributions for 13C6-glucose are shown in Appendix A.

**Figure 3 cancers-11-01488-f003:**
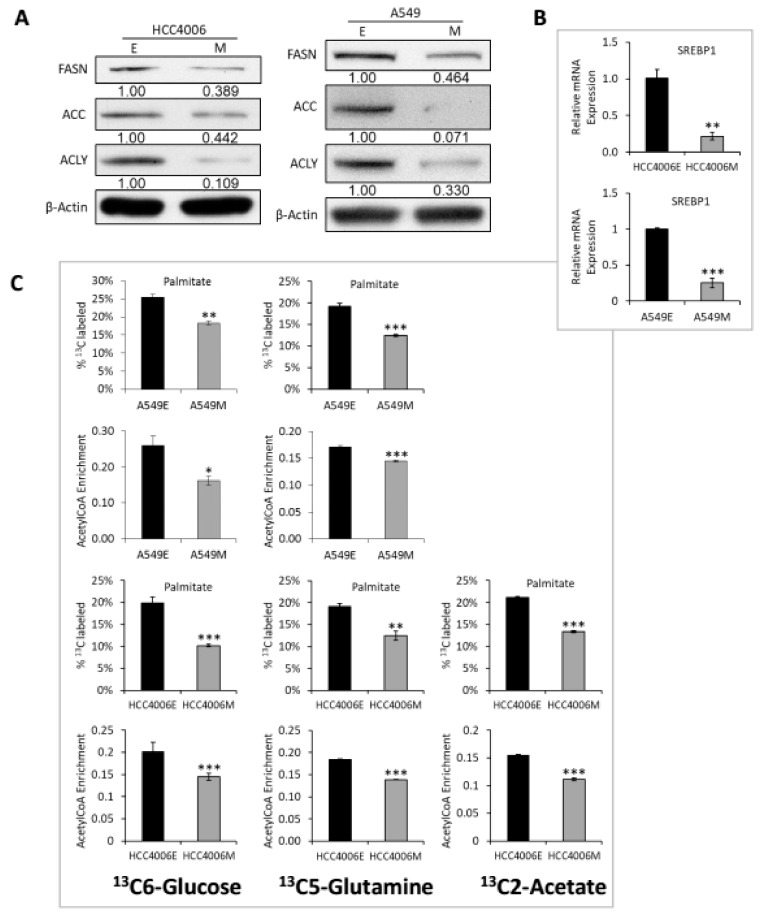
De novo lipid synthesis is decreased in isogenic mesenchymal cell states. (**A**) In HCC4006 and A549 cells, lipogenic enzymes decrease in the M state by immunoblot. (**B**) SREBP1/SREBF1 RNA is reduced in the M state (*p* < 0.05 *; *p* < 0.01 **; *p* < 0.001 ***). (**C**) A549 and HCC4006 have decreased incorporation of 13C6-glucose, 13C5-glutamine, and/or 13C2-acetate into palmitate in the M state. They also show decreased acetyl-CoA enrichment by their respective substrates. Isotopologue distributions are shown in Appendix A.

**Figure 4 cancers-11-01488-f004:**
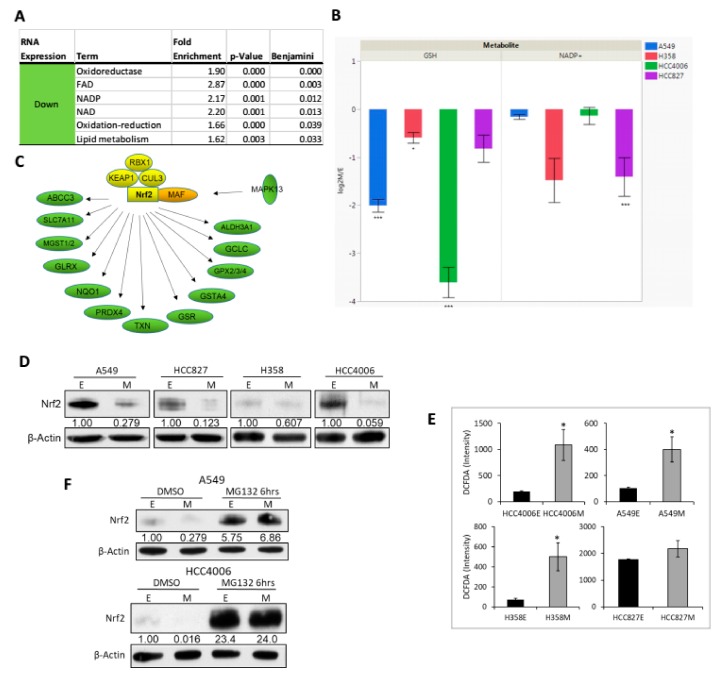
Wild type and mutant KEAP1 mesenchymal state cells show attenuated Nrf2 protein and activity. (**A**) Enrichment of down regulated redox and lipid metabolic genes correlated with M state (DAVID). (**B**) Reduced abundance of glutathione and NADP+ as measured by LC-MS/MS with log2 fold change M/E (*p* < 0.05 *; *p* < 0.01 **; *p* < 0.001 ***). (**C**) Down regulation (green) of Nrf2/NFE2L2 target genes in the mesenchymal state supports reduced Nrf2 activation in the mesenchymal state. Nrf2 associated transcription factor MAF was markedly increased (orange), with little/no change in Nrf2 protein stability regulators KEAP1, CUL3, and RBX1 (yellow). (**D**) Nrf2 levels decrease in the M state at the protein level in all four EMT models. (**E**) Reduced Nrf2 protein in M state cells correlates with increased ROS as measured by flow cytometry. (**F**) Both E and M state cells turnover Nrf2 protein via proteasomal degradation. A549 and HCC4006 were treated with proteasome inhibitor MG132 for 6 h followed by immunoblot for Nrf2 and β-actin.

**Figure 5 cancers-11-01488-f005:**
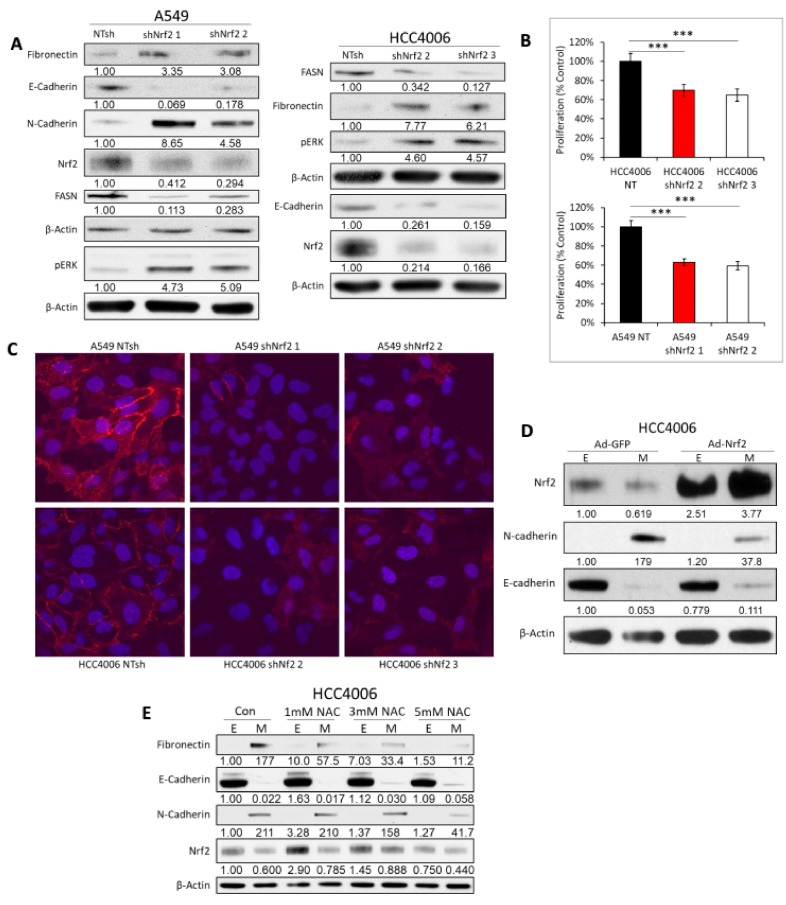
Nrf2 promotes an epithelial phenotype and antagonizes the mesenchymal state. (**A**) Nrf2 knockdown in A549 and HCC4006 leads to an increase in mesenchymal marker expression and a decrease in epithelial marker expression, as well as a decrease in FASN and increase in pERK by immunoblot. (**B**) Nrf2 knockdown in A549 and HCC4006 decreases cell proliferation over 72 h (*p* < 0.05 *; *p* < 0.01 **; *p* < 0.001 ***). (**C**) Nrf2 Knockdown in A549 and HCC4006 causes loss of E-cadherin (red) from the cell membrane by IF, where nuclei are stained with DAPI (blue). (**D**) When Nrf2 is overexpressed in HCC4006, mesenchymal marker expression decreases (N-cad) while the epithelial marker (E-cad) increases by immunoblot. (**E**) In HCC4006, treatment with 1, 3, 5 mM NAC over the course of 72 h, a loss of M markers and gain of E markers is seen in the M state.

**Figure 6 cancers-11-01488-f006:**
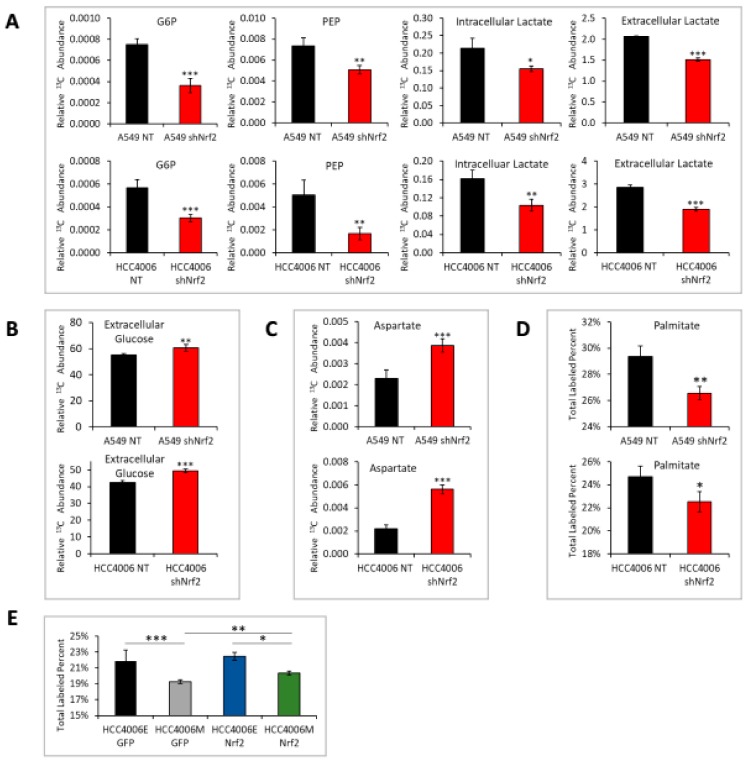
Knockdown of Nrf2 phenocopies mesenchymal state metabolic alterations. (**A**) In HCC4006 and A549 with Nrf2 knockdown treated with 13C glucose; production of G6P, PEP, intracellular lactate, and extracellular lactate are reduced (*p* < 0.05 *; *p* < 0.01 **; *p* < 0.001 ***). (**B**) Nrf2 knockdown cells have increased extracellular glucose. (**C**) Nrf2 knockdown increases aspartate. (**D**) With Nrf2 knockdown, incorporation of 13C6-glucose into palmitate is attenuated. Isotopologue distributions are shown in Appendix A. (**E**) Conversely, when Nrf2 is overexpressed in HCC4006, incorporation of 13C6-glucose into palmitate is increased (GFP control vs. Nrf2; *p* < 0.01).

**Figure 7 cancers-11-01488-f007:**
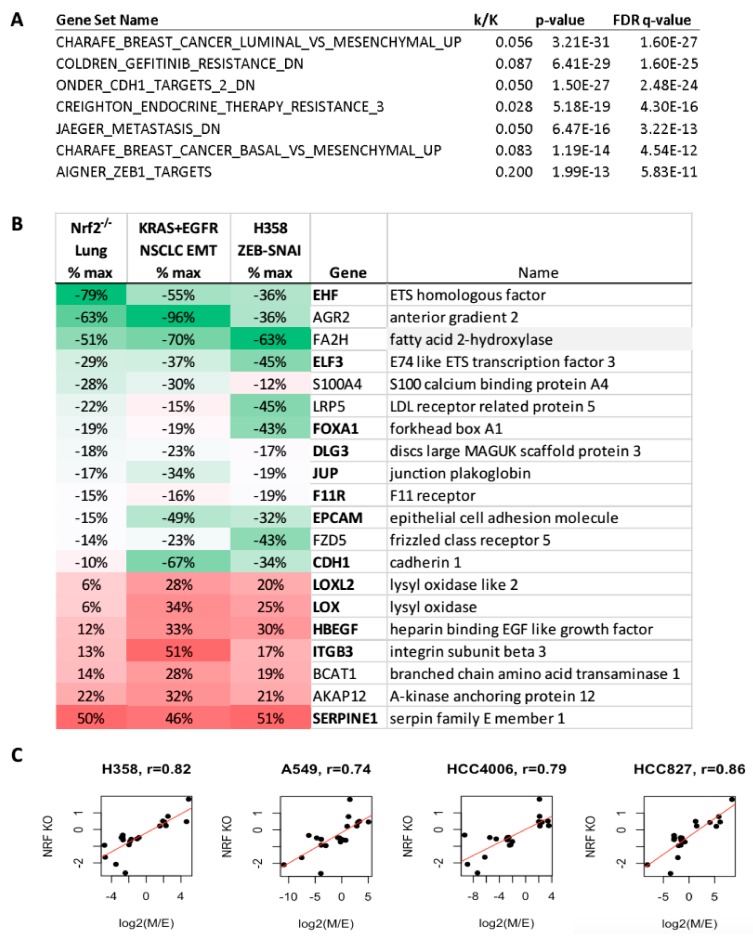
Mouse Nrf2^−/−^ promotes a mesenchymal phenotype in primary lung alveolar cells. (**A**) Gene set enrichment of Nrf2 KO RNA expression indicate correlation with multiple EMT signatures. (**B**) Lung alveolar cell RNA expression comparing Nrf2 knockout vs. control (column 1), is compared with mean log2 TGFβ E and M state changes in the four lung adenocarcinoma cell models (column 2) and mean Zeb1 and Snail1 driven EMT changes in two H358 models. Data are expressed as percent maximum change to scale microarray and RNAseq dynamic ranges. EMT related RNAs are bolded. (**C**) Correlation of Nrf2 KO RNA expression with EMT RNA changes in the four NSCLC models.

**Figure 8 cancers-11-01488-f008:**
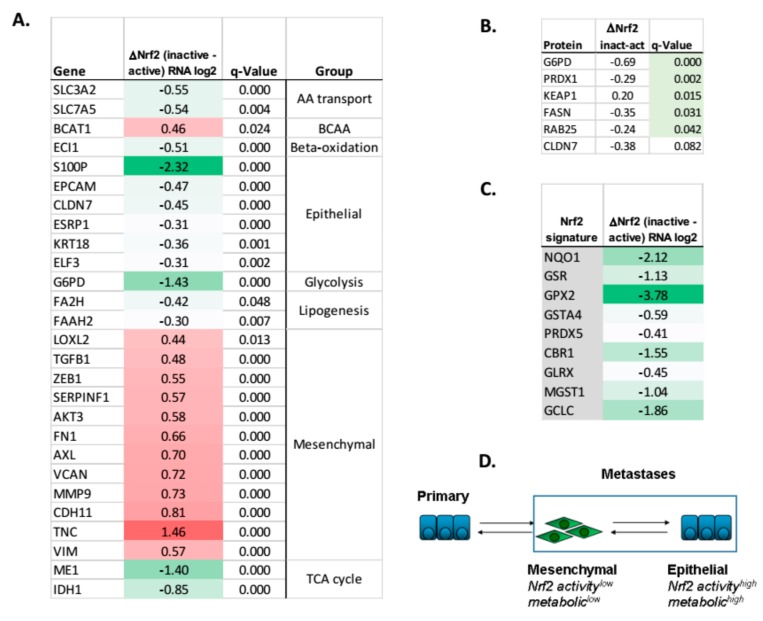
Attenuation of Nrf2-dependent transcription correlates with EMT and metabolic RNA and protein abundance in NSCLC adenocarcinomas. (**A**) Co-correlation of Nrf2 target RNA expression (anchor genes NQO1, GSR, GPX2, GSTA4, PRDX5, CBR1, GLRX, MGST1, GCLC) with a reference 85 gene EMT signature defined [60]. The TCGA Provisional 503 patient lung adenocarcinoma patient dataset was queried. Nrf2 target RNA expression was positively correlated with an epithelial state. Nrf2 activation correctly correlated with KEAP1 mutation (q = 2.4 ×10^−4^). (**B**) Decreased G6PD and FASN metabolic proteins and decreased RAB25 and CLDN7 epithelial proteins, when Nrf2 is inactive (or KEAP1 expression is elevated) and target gene RNAs (e.g., PRDX1) are reduced. (**C**) Nrf2 signature RNA are coordinately expressed between inactive (signature low) and active (signature high) TCGA specimens. Nrf2 signature RNA are coordinately regulated between inactive (signature low) and active (signature high) TCGA specimens. (**D**) Model of Nrf2 and metabolic activity in metastatic lung adenocarcinoma where Nrf2 bifurcates into Nrf2/metabolic^low^ and Nrf2/metabolic^high^ in mesenchymal and epithelial cell states, respectively.

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
