# Peer review of "Decoupling of Nrf2 Expression Promotes Mesenchymal State Maintenance in Non-Small Cell Lung Cancer"

_cancers, 2019, doi:10.3390/cancers11101488_

Round 1

Reviewer 1 Report

The authors provided evidence that reduction of Nrf2 expression may increase ROS accumulation in prometastatic mesehchymal cells. Conversely, overexpression of Nrf2 promoted epithelial phenotype and reduced glycolytic, TCA cycle, andlipogenic output from both glucose and glutamine in the KRAS or EGFR-mutated lung cancer cells. The authors concluded that altering Nrf2 expression may modulate epithelial/mesenchymal cancer cell transition, with implication for therapeutic inhibition of Nrf2 in cancer treatment. However, similar findings have been reported previously (Zhou et al., 2016, 6:38646, Sci Rep; Bocci et al., 2019, 11:251-263, Integrative Biol). The title of the manuscript should be changed to "Decoupling of Nrf2 expression promotes----" not "activity". It is well known that Nrf2 plays a dual role in lung tumorignesis. However, the underlying mechanism of Nrf2 reduction on the formation of EMT did not investigate. 

Author Response

We thank the reviewers for their comments. Black text reflect comments from the reviewers (italic is overall; specific questions in bold). Blue text reflects the author’s response. Green text reflects new text inserted in the manuscript, with existing surrounding text (for context) in brown.

Reviewer 1 commented:

The authors provided evidence that reduction of Nrf2 expression (via EMT) may increase ROS accumulation in pro-metastatic mesenchymal cells. Conversely, overexpression of Nrf2 promoted epithelial phenotype and increased glycolytic, TCA cycle, and lipogenic output from both glucose and glutamine in the KRAS or EGFR-mutated lung cancer cells. The authors concluded that altering Nrf2 expression may modulate epithelial/mesenchymal cancer cell transition, with implication for therapeutic inhibition of Nrf2 in cancer treatment.

 The Introduction and Discussion should contain more information:

We included the information of Nrf2 regulation in the Introduction section:

Growth factor activation and mutations in KRAS increase Nrf2 transcription, in part through AP1, and Nrf2 expression, which correlate with tumor viability, proliferation [34] and mRNA translation [35]. Nrf2 protein abundance is carefully regulated by ubiquitination through redox state sensitive E3 ligase complexes including KEAP1/CUL3, BTRC/SKP1 and SYVN1 [36], as well through DNA mutation [37] and through alternative RNA splicing. Nrf2 activating mutations in KEAP1, CUL3 or Nrf2 itself increase cell proliferation and tumor expansion in vivo [38, 39] and can contribute to drug resistance [40]. In normal cells, Nrf2 can exert a geno-protective function by preventing excess reactive oxygen species (ROS), while in tumor cells Nrf2 activation contributes to tumorigenesis [34, 38, 39, 41].

We also included more information on EMT and cancer metastasis in the Introduction section:

EMT-derived cells can exhibit stem-like properties [16] that include the ability to initiate tumors at low cell density in vivo, sphere formation in vitro and expression of potential pluripotent stem cell associated markers (i.e., CD44high, CD24low, ALDH1) that likely reflect the underlying chromatin remodeling events. Distinctively, later stage lung tumors can exhibit nests of stable mesenchymal sarcomatoid-like cells [17] associated with aggressive proliferative and migratory behaviors. Current evidence show that mesenchymal-like tumor cells can be positively selected for by chemotherapy in solid tumors [18] and resist apoptosis in part through suppression of BIM [19]. The effective therapeutic targeting of these reversible /metastable and sarcomatoid cancer states is an unmet medical need [20, 21], in part due the high degree of cellular heterogeneity [22] and the acquisition of parallel signaling pathways within individual cells.

Proliferating tumor cells can increase glycolytic, lipid and nucleic acid synthesis through the Warburg effect. However, both decreased and increased glycolysis and oxidative phosphorylation have been correlated with cancer metastasis, seemingly in a model specific manner. EMT has been associated with decreased lipogenesis and glycolysis [36] but this also can vary between cell models.

We included more detail around the conclusions and their bearing on published knowledge on EMT, Nrf2 and metabolism as indicated below (A).

Similar findings have been reported previously (Zhou et al., 2016, 6:38646, Sci Rep; Bocci et al., 2019, 11:251-263, Integrative Biol).

The role for Nrf2 expression in cancer progression has yielded divergent findings in different model systems. For example, Zhou et al. examined Nrf2 function in fibrotic lung disease where Nrf2 expression decreased the EMT promoting transcription factor Snail. In contrast, Bocci et al. found Nrf2 knockout in bladder RT4 downregulated the epithelial marker E-cadherin as well as the mesenchymal marker Zeb1, through generation of an intermediate ‘hybrid’ EMT. Bocci et al also find in NSCLC H1975 cells that one of two Nrf2 siRNAs stimulates EMT. We point out the H1975 have a substantial population of cells that have already undergone EMT (Argast et al., 2011; Clin. Expt. Metastasis), where cell selection can be a confounding factor. The studies of Zhou and Bocci have different conclusions with respect to Nrf2 and EMT, where Nrf2 either inhibits or stimulates EMT respectively.

Here in multiple mutant KRas and mutant EGFR NSCLC models we find Nrf2 expression promotes a more epithelial cell state while antagonizing the mesenchymal state, with little observation of a hybrid state.

We have included this in the Discussion section:

The role for Nrf2 expression in cancer progression has yielded divergent findings in different model systems. For example in fibrotic lung disease, Nrf2 expression decreased the EMT promoting transcription factor Snail [63]. In contrast, in NSCLC H1975 and bladder RT4 cells Nrf2 knockdown had mixed impacts on epithelial and mesenchymal markers, suggesting the generation of an intermediate ‘hybrid’ EMT state [64]. Here in mutant KRas and mutant EGFR NSCLC models we find Nrf2 expression promotes a more epithelial cell state while antagonizing the mesenchymal state. Nrf2 is known to play a dual role in cancer, where Nrf2 is understood to reduce genotoxic damage in normal cells, yet promote a proliferative behavior in established epithelial-derived carcinoma cells. Here we uncover different functions of Nrf2 in carcinoma’s dependent on epithelial or mesenchymal EMT cell state, a third role.

The title of the manuscript should be changed to "Decoupling of Nrf2 expression promotes----" not "activity".

We have made this change as requested.

It is well known that Nrf2 plays a dual role in lung tumorigenesis.

This text was included in the Discussion:

Nrf2 is known to play a dual role in cancer, where Nrf2 is understood to reduce genotoxic damage in normal cells, yet promote a proliferative behavior in established epithelial-derived carcinoma cells. Here we uncover different functions of Nrf2 in carcinoma’s dependent on epithelial or mesenchymal EMT cell state, a third role.

However, the underlying mechanism of Nrf2 reduction on the formation of EMT did not investigate.

We have investigated the role of protein turnover, through proteasomal mechanisms in part independent of Keap1 in the regulation of Nrf2 levels.  More specifically we describe non-KEAP1 regulation of Nrf2 in A549 cells. This text is in the Results section: The decreased abundance of Nrf2 protein was associated with increased ROS (Figure 4E). To further investigate the regulation of Nrf2 levels in epithelial and mesenchymal states, we measured the stability of Nrf2 after both proteasome inhibition and ribosomal translation blockade. Nrf2 protein abundance was measured after proteasome inhibition with MG132 (10 µM) for six hours in epithelial and mesenchymal, A549 and HCC4006 backgrounds (Figure 4F). Control cells showed reduced Nrf2 in the mesenchymal state. By six hours, proteasomal inhibition resulted in a marked increase in Nrf2 protein to similar levels in both cell states. These data indicate that Nrf2 is degraded in part through a proteasomal pathway in both mt-KEAP1 (A549) and wt-KEAP1 (HCC4006). To determine the effect of the mesenchymal state on de novo Nrf2 protein synthesis we treated A549 and HCC4006 cells in both states with cycloheximide (10 µM) for one and three hours (Supplementary Figure S4E). Nrf2 protein was rapidly reduced in both epithelial and mesenchymal states and in mutant (A549) and wild type (HCC4006) KEAP1 backgrounds. The decrease in Nrf2 protein and Nrf2 activity and increase in ROS suggest EMT alters the stability of Nrf2, in part independent of KEAP1, and that the increase in ROS might be a result of decreased Nrf2. 

We do suggest several approaches further investigating alternative E3-ligases, alternative Nrf2 splice forms and altered translational efficiency beyond the scope of this study, focusing more on the relationships between Nrf2, EMT and metabolism. In the Discussion: This is a point of future investigation, examining alternative splice forms of Nrf2, alternative ubiquitin E3 ligases which might recognize Nrf2 as substrate and translational control of Nrf2 synthesis.

Reviewer 2 Report

Re: Review of the manuscript entitled “ Decoupling of Nrf2 activity promotes mesenchymal 2 state maintenance in non-small cell lung cancer”.

Dear Editor

Please find enclosed review of the manuscript (reference is Cancers-595831)

In the present work, Jaley  et al. have found and suggested the role of Nrf2, which is a key transcription factor controlling the expression of redox regulators, in the epithelia mesenchymal transition (EMT) and metabolic alterations in non-small cell lung cancer (NSCLC) using various in vitro and in vivo approach. Knockdown of Nrf2 promoted a mesenchymal phenotype and reduced glycolytic, TCA cycle and lipogenic output from both glucose and glutamine in the isogenic cell models; while overexpression of Nrf2 promoted a more epithelial phenotype and metabolic reactivation.  Knockdown of Nrf2 promoted a mesenchymal phenotype and reduced glycolytic, TCA cycle and lipogenic output from both glucose and glutamine in the isogenic cell models; while overexpression of Nrf2 promoted a more epithelial phenotype and metabolic reactivation. 

In normal cells, Nrf2 can exert a geno-protective function by preventing excess reactive oxygen species (ROS), while in tumor cells Nrf2 activation contributes to tumorigenesis. However, the conflict exist regarding the relation Nrf2 activity and EMT, which may be dependent on cell type or genetic backgroud, etc.   

The results showed in the paper is very interesting that the role of Nrf2 activity in the process (or it can be specific step) of EMT.

There is no specific issue to considered  except  page 8, line 229 2.4. rf2 activity and protein abundance are regulated by EMT state need to be changed Nrf2.

Author Response

We thank the reviewers for their comments. Black text reflect comments from the reviewers (italic is overall; specific questions in bold). Blue text reflects the author’s response. Green text reflects new text inserted in the manuscript, with existing surrounding text (for context) in brown.

Reviewer 2 commented:

In the present work, Haley  et al. have found and suggested the role of Nrf2, which is a key transcription factor controlling the expression of redox regulators, in the epithelia mesenchymal transition (EMT) and metabolic alterations in non-small cell lung cancer (NSCLC) using various in vitro and in vivo approach. Knockdown of Nrf2 promoted a mesenchymal phenotype and reduced glycolytic, TCA cycle and lipogenic output from both glucose and glutamine in the isogenic cell models; while overexpression of Nrf2 promoted a more epithelial phenotype and metabolic reactivation.  Knockdown of Nrf2 promoted a mesenchymal phenotype and reduced glycolytic, TCA cycle and lipogenic output from both glucose and glutamine in the isogenic cell models; while overexpression of Nrf2 promoted a more epithelial phenotype and metabolic reactivation.  In normal cells, Nrf2 can exert a geno-protective function by preventing excess reactive oxygen species (ROS), while in tumor cells Nrf2 activation contributes to tumorigenesis. However, the conflict exist regarding the relation Nrf2 activity and EMT, which may be dependent on cell type or genetic background, etc.  The results showed in the paper is very interesting that the role of Nrf2 activity in the process (or it can be specific step) of EMT.

There is no specific issue to considered except page 8, line 229 2.4. rf2 activity and protein abundance are regulated by EMT state need to be changed Nrf2.

This typo was corrected.

Reviewer 3 Report

Summary: The authors use an in vitro EMT model to understand changes in metabolism. They report that both glycolysis and TCA cycle metabolism are lower in cells in the "metastable" state. This corresponded with reduced NRF2 expression and knock down of NRF2 showed a similar phenotype with lowered glycolysis and TCA cycle metabolism. Nrf2 knockout mouse lung alveolar cells showed mesenchymal signatures. Finally, the authors confirm some of these findings in human NSCLC adenocarcinomas. Overall, the authors suggest that NRF2 may regulate the mesenchymal state in lung cancers.

Comments: Overall the manuscript is well written, illustrated and supported by data confirmed in multiple cell lines. The following major comments are noted to strengthen the conclusions that could be drawn from the manuscripts.

In figure 6E the authors show a modest increase in 13C-glucose incorporation into palmitate when NRF2 is overexpressed. Do glycolytic and TCA cycle metabolites increase in this condition? In figure 7, the authors suggest an increase in mesenchymal signatures in Nrf2 knockout mouse lung alveolar cells. Do glycolytic and TCA cycle genes change here? From the TCGA data in figure 8, are there any clinical differences (such as overall survival, type of adenocarcinoma or response to treatment) between  Nrf2/metabolic low 3and Nrf2/metabolic high samples?

Author Response

We thank the reviewers for their comments. Black text reflect comments from the reviewers (italic is overall; specific questions in bold). Blue text reflects the author’s response. Green text reflects new text inserted in the manuscript, with existing surrounding text (for context) in brown.

Reviewer 3 commented:

Summary: The authors use an in vitro EMT model to understand changes in metabolism. They report that both glycolysis and TCA cycle metabolism are lower in cells in the "metastable" state. This corresponded with reduced NRF2 expression and knock down of NRF2 showed a similar phenotype with lowered glycolysis and TCA cycle metabolism. Nrf2 knockout mouse lung alveolar cells showed mesenchymal signatures. Finally, the authors confirm some of these findings in human NSCLC adenocarcinomas. Overall, the authors suggest that NRF2 may regulate the mesenchymal state in lung cancers.

Comments: Overall the manuscript is well written, illustrated and supported by data confirmed in multiple cell lines. The following major comments are noted to strengthen the conclusions that could be drawn from the manuscripts.

 In figure 6E the authors show a modest increase in 13C-glucose incorporation into palmitate when NRF2 is overexpressed. Do glycolytic and TCA cycle metabolites increase in this condition?

With adenovirus expression of Nrf2 we do not observe coherent changes in glycolysis or TCA cycle intermediates by 48 hours, a relatively early time during the EMT transition. At later time points adenovirus infection can compromise cell viability. Lipogenesis has been a more robust co-correlate of EMT and metabolism and which may facilitate its detection here.

In figure 7, the authors suggest an increase in mesenchymal signatures in Nrf2 knockout mouse lung alveolar cells. Do glycolytic and TCA cycle genes change here?

In the Affymetrix U133Plus2 data from Nrf2-/- alveolar cells, Gene Set Enrichment (GSEA) analysis and manual inspection revealed no coherent RNA transcript changes in glycolytic or TCA cycle encoded genes. The EMT phenotype was the dominant GSEA feature in this Nrf2 KO dataset.

From the TCGA data in figure 8, are there any clinical differences (such as overall survival, type of adenocarcinoma or response to treatment) between Nrf2/metabolic low 3and Nrf2/metabolic high samples?

We do observe a trend for reduced overall survival in patients with high Nrf2 signature expression (p=0.15) using the most recent TCGA Provisional lung adenocarcinoma 586 sample set.  We have included this data in the Supplementary Figure S9 and included this in the text:

Analysis of breast cancer patient samples for high Nrf2 expression have correlated with a significant decrease in overall survival (p=0.024) [64]. Here in NSCLC adenocarcinomas we observed a trend for poor overall survival with increased Nrf2 target gene expression (p=0.15), which is correlated with a proliferative, epithelial phenotype. Taken together, these data support a model (Figure 8D) where EMT promotes dissemination of slowly proliferating migratory cells with low rates of glycolytic and lipogenic output, where the reestablishment of macro metastases at distant sites requires MET, and an epithelial proliferative phenotype with increased metabolic output.

Round 2

Reviewer 1 Report

The revised manuscript is sufficient to response the questions of this reviewer.  

Reviewer 3 Report

The authors have addressed all concerns